# Optimal Design and Verification of Informal Learning Spaces (ILS) in Chinese Universities Based on Visual Perception Analysis

**Yuzhen Chen** [1,2], **Jinxiu Wu** [1,*], **Yamei Zou** [1], **Wei Dong** [1] and **Xin Zhou** [1]

1 School of Architecture, Southeast University, Nanjing 210096, China
2 School of Architecture, Xiamen University Tan Kah Kee College, Zhangzhou 363105, China
* Correspondence: wu_jinxiu@seu.edu.cn; Tel.: +86-138-5167-3898

**Abstract:** As the focus on higher education in China gradually shifts from rapid development to an emphasis on quality, the need for campus environments to become facilitators of education has gained increasing attention. The accelerated development of information technology has also led to tremendous changes in both teaching and learning methods, with informal learning taking on an increasingly important role. Furthermore, the development of human sensing technology, especially visual perception technology, has brought in new opportunities for the research and optimization of informal learning spaces (ILSs) in universities. This paper focuses on the ILS in Chinese universities by exploring optimal design approaches based on visual perception analysis. Through research and field investigation, this paper proposes revised theoretical research of classifications and spatial elements of ILS in universities more applicable to the architectural study of space. This paper also explores practical optimal design methods with two case studies and makes experiments with wearable eye trackers to study the users' perception in these spaces before and after optimization. The optimal design is made from the aspects of physical space, facilities, and environment. Visual perception experiments and quantitative analysis were used to obtain a higher level of experimental accuracy than the previous studies and thus to study the real feeling of users in spaces. By these means, the effect of the optimized design was verified and the relation between users' perceptions and the spatial environments was explored for further improvements to optimal design methods. This article can provide theoretical and practical references for campus space optimization research and design, especially for ILS on university campuses.

**Keywords:** informal learning space (ILS); visual perception analysis; wearable eye tracker; optimal design and verification

## 1. Introduction

The accelerated development of China's economy after the reform has provided unprecedented opportunities for the development of universities. In 2019, the number of enrolled students in China's universities reached 38.33 million, which ranked it first in the world [1]. The number of universities was 2824, ranking second in the world. The development of Chinese higher education has entered a new stage with the development model shifting from incremental development to stock optimization [2]. In the new stage, informal learning space (ILS) becomes as crucial as the teaching space since Chinese universities are paying more attention to the cultivation of students' comprehensive ability. While the way of cultivation is no longer limited to teaching in classrooms; dialogue, seminars, communication, contemplation, and even rest have become significant ways to enhance students' abilities. However, research shows that both the quantity and quality of ILS in Chinese universities need to be improved [3–8].

Research on ILS began with a qualitative discussion at a theoretical level, covering the influencing factors, student behaviors, and their satisfaction with the space. For example,

Ahamd A. Alhusban assessed students' needs for ILS through questionnaires and studied ways to meet these needs through corresponding guidelines [9]. Descriptive statistics and Pearson correlation coefficient was applied to analyze the data, and the results revealed that interrelationships among all campus urban design principles have a strong positive linear relationship. Moreover, Xianfeng Wu studied the relationship between ILS design points and behavior patterns through field research and questionnaires [3]. Six significant design characteristics influencing the use of ILS was identified in this study, including comfort, flexibility, functionality, spatial hierarchy, openness, and other support facilities. Shirley Dugdale [10] proposed an "informal learning landscape" for campus planning by integrating both formal and informal learning spaces. He highlighted the criticality of learning activities and human interaction and suggested that campuses need to create a participatory architecture to support these communities of learners. Maheran et al. [11] investigated how ILS on university campus outdoor spaces can enhance students' academic performance. This study suggested that it is important to take the outdoor classroom into consideration during the campus landscape design, as it affects students' activities such as learning, educating, meeting, exploring, as well as relaxing. Ran Ranjin et al. [12] investigated the relevance of informal learning behaviors to campus open spaces through field behavioral observations and provided recommendations for space regeneration design, highlighting the importance of informal learning behaviors in the design of spatial regeneration. Tian Yang [13] explored the relationship among "Information, space, and people" in research on three types of ILS in college libraries in the United States. This study emphasized the construction of diverse learner-centered spaces. Among them, the most famous research is Lennie Scott-Webber's research theoretical framework of ILS and its design guidelines [14]. She stressed that as higher education undergoes dramatic changes due to changes in pedagogy and technology, which are reshaping the needs of learners, the planning, design, and use of learning spaces must change. Therefore, previous studies have carried out some theoretical research on the optimal design of ILS, and several research have concluded that it is important to strengthen the interaction between ILS and learners, emphasizing the importance of the human experience created by ILS to the design effect. However, there is a lack of ILS classification methods applicable to architecture space research, and few studies have focused in-depth research on the relationship between the spatial elements of ILS and users' feelings on it in a quantitative way. This affects the application of the theories and methods in the practice design of ILS.

On the other hand, the rapid development of new technologies has contributed significantly to the depth of human perception research [15–18]. Among them, visual perception, as an essential part of human perception, is one of the most important ways in which humans perceive space. For example, ETH Zurich optimized the signage design and space organization in Frankfurt Airport with help of data collected from subjects wearing eye-tracking devices [19]. Dr. Nikhil Naik et al. at MIT used "Streetchange", a machine learning technology that provides a visual approach to urban space design and street image research to assess and analyze the elements in urban space that affect perception [20]. Dr. Alexander Erath et al. from the Future Cities Lab at ETH Zurich built a highly realistic virtual reality environment by combining 3D modeling and traffic simulation techniques. Although most of the previous literature presents the use of new technologies of visual perception in urban public spaces and streets, the study by Lebrun, C. points out that people's visual perception of architectural space is not related to the type of building, but depends mainly on landscape elements [21]. In addition, Małgorzata Lisi' nska-Ku´snierz's research shows the applicability of eye-tracking in assessing the visual perception of architectural works and points to its potential for the disciplines of architecture and urban planning [22]. The eye-tracking technology helps to accurately establish the relationship between the user's visual perception in space and the spatial elements so that the spatial components of ILS can be studied and optimized accordingly to promote its use. Therefore, it is necessary to introduce human perception technologies into the campus ILS research.

Different from traditional theoretical studies of ILS, the above-mentioned research has brought new opportunities for the design of ILS, especially for the interaction between users. Some researchers have tried to apply new technological tools (visual perception) in urban public areas and streets [23,24]; however, little research has been applied to the optimal design of ILS. Therefore, based on the problems found in the extensive investigation and literature review, this paper proposes revised theoretical research for classifications and spatial elements of campus ILS more applicable to architecture space study. This paper also explores practical optimal design methods for campus ILS and verifies the effect of optimal design with the help of new technologies. The visual perception technology represented by the wearable eye-tracking device is introduced to collect data on users' feelings in ILS at a much higher level of accuracy than in previous studies and thus improve the quality and efficiency of optimal design. The methodology in this paper can provide theoretical and practical references also for other campus space research.

This study aims to investigate the optimal design and verification of ILS in Chinese universities using visual perception analysis to solve existing problems and promote usage by both faculty and staff.

The objectives of this study are as follows:

- To verify the effect of optimal design of ILS on visual perception experiment and quantitative analysis.
- To study the relationship between users' visual perception and spatial elements for further optimal design methods.
- To propose practical recommendations to enhance the use of ILS in Chinese universities.

## 2. Methodology

Based on the above objectives, the framework of this research is as follows: Firstly, a theoretical analysis of classifications and spatial elements of ILS was presented through research and investigation. Secondly, two case studies are made to illustrate the research and optimal design methods. Eye-tracking data before and after the optimal design were obtained through visual perception experiments [25], and quantitative analyses were conducted to study the relationship between users' visual perception and spatial elements and thus verify the design effect. Finally, the optimal design method for ILS in Chinese universities is summarized and enhanced.

The research framework of this study is shown in Figure 1.

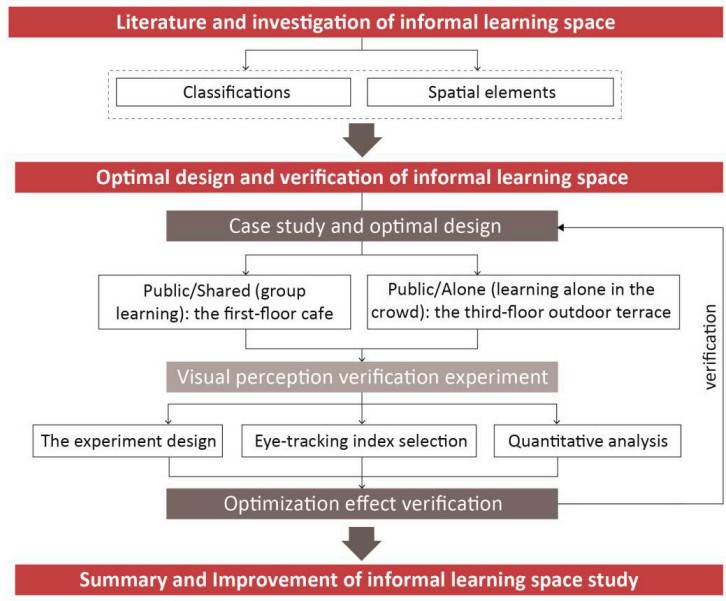

**Figure 1.** Framework of the research.

### 2.1. Litterateurs and Investigation

#### 2.1.1. Litterateurs and Classification

While U.S. higher education had impressive success in its nearly 400-year history, the planning and design of learning spaces on U.S. campuses also reflect a more programmatic approach to formal learning and learning. As today's teaching methods and campus spaces are changing, variation in learning spaces is inevitable. For this reason, Weber proposed a four-quadrant theoretical framework to support those learning behaviors in informal learning environments. The quadrant theory framework consists of four quadrants, Private/Alone, Private/Together, Public/Alone, and Public/Together, as shown in Figure 2a [8]. The advantage of this theory is that it covers all types of behaviors and spaces that occur in ILS. Nevertheless, the drawback of this theory is that it does not reflect the ambiguous and transitional ILS. Based on the problems that emerged from this theory and preliminary study, this paper further broadens and optimizes the theoretical study to make it more applicable to the study of architectural space.

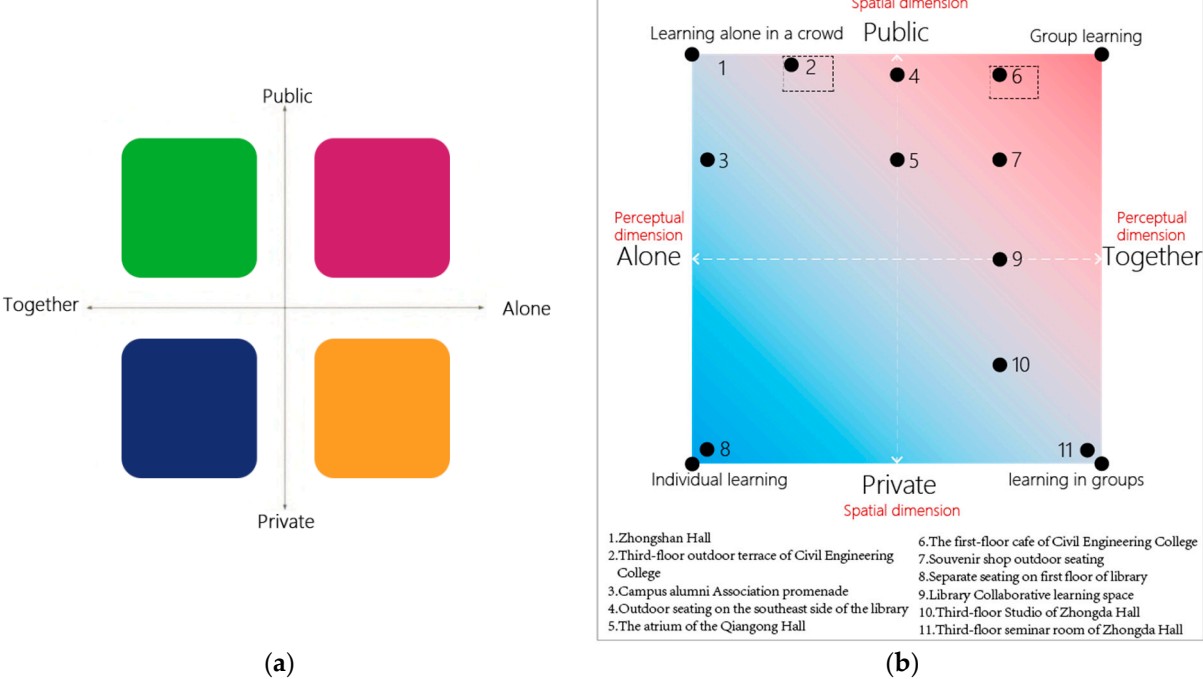

**Figure 2.** Classification of ILS in universities. (**a**) Weber's four-quadrant theoretical framework.; (**b**) The framework for classifying ILS in this paper and the 11 ILS investigated in Southeast University.

This paper suggests a revised theoretical framework to support engaged learning behaviors by dividing ILS into four categories from the spatial and perceptual dimensions as the vertical and horizontal coordinates of the four-quadrant framework, which are Private/Alone (Individual learning), Private/Shared (Group learning), Public/Alone (Learning in the crowd), Public/Shared (Group learning), and Public/Alone (Open group learning).

(1) Private/Alone (Individual learning): closed and smaller private spaces with defined boundaries, usually found in the library, with students' static behaviors.

(2) Private/ Shared (Learning in groups): private spaces with defined boundaries and moderate scale, usually located inside academic buildings or libraries, with student groups' relatively dynamic learning behaviors.

(3) Public/Alone (Learning alone in the crowd): open spaces usually located in public areas, such as study rooms in libraries, gray spaces, etc., with students learning behaviors changing according to the atmosphere of the surroundings.

(4)    Public/Shared (Group learning): open and more functionally complex spaces usually located in cafes, normally with student groups' dynamic learning behaviors, and some of them also connect with outdoor spaces.

2.1.2. Field Investigation

An extensive study of ILS in universities was conducted in Nanjing, with a total of 61 cases of field investigation on 11 campuses. Considering the representativeness of the case study and the convenience of subsequent experiments, this paper focuses on the Sipailou Campus of Southeast University for an in-depth study. The reasons were as follows:

Firstly, Southeast University was established in 1902 and is one of the earliest and most prestigious universities in China, and Sipailou Campus is the oldest campus of Southeast University. During its 120 years of development, it has formed a campus space and architecture with both a long history and contemporary development and has a strong representation of almost the entire process of the construction and development of university campuses in modern China.

Secondly, the School of Architecture of Southeast University, which is located on this campus, is one of the top three architecture schools in China and has the top experimental facilities in the field of architecture, and the advanced experimental conditions are conducive to the subsequent experiments and research.

A total of 11 cases of ILS in Sipailou Campus, Southeast University were studied and classified in Figure 2b. Then, case studies and optimal design were made for two spaces: the first-floor cafe (Public/Shared) and the third-floor terrace (Public/Alone) in the building of the School of Civil Engineering of Southeast University. These studies aim to cover the most frequent types of public ILS in the investigation and contain both indoor and outdoor ILS.

2.1.3. Spatial Elements Study

In field investigation, it is shown that ILSs in universities are composed of three elements: physical space, facilities, and environment. The former two can be directly perceived through visuals, while the latter can be mainly perceived by visuals and/or feelings.

(1)    Physical Space: the material elements of space, including the location, size, enclosure, material, color, etc.
(2)    Facilities: the elements to help with space functions, often in the form of furniture and equipment, and linked with users' behavior.
(3)    Environment: Including landscape and the physical environment. The landscape environment emphasizes the natural and artificial environments both indoor and outdoor. The physical environment refers to the thermal, optical, and acoustic environment, etc., influenced by material elements. Environment analysis based on visual perception in this study involves only the landscape environment.

*2.2. Case Studies and Optimal Design*

2.2.1. Case Studies

Two spaces were chosen in this paper to make case studies in the practice project of optimal design of the College of Civil Engineering building of Southeast University. They were the first-floor cafe as the typical case for the "Public/Shared" type and the third-floor outdoor terrace for the "Public/Alone" type. The reasons were follows: First, it was discovered in former investigations that the number of public ILS in the universities was much more than that of private ones and was used much more frequently. Second, both interior and exterior ILS should be covered.

(1)    The first-floor cafe (Public/Shared):

The cafe on the first floor was optimized from the original activity center and the adjacent sub-foyer, as shown in the red area in Figure 3. Before optimization, the original

activity center was separated from the sub-foyer by a wall. Investigation showed that the activity center activity was used frequently, yet the sub-foyer was always empty. The main problem with this space is the lack of facilities for people to stay in.

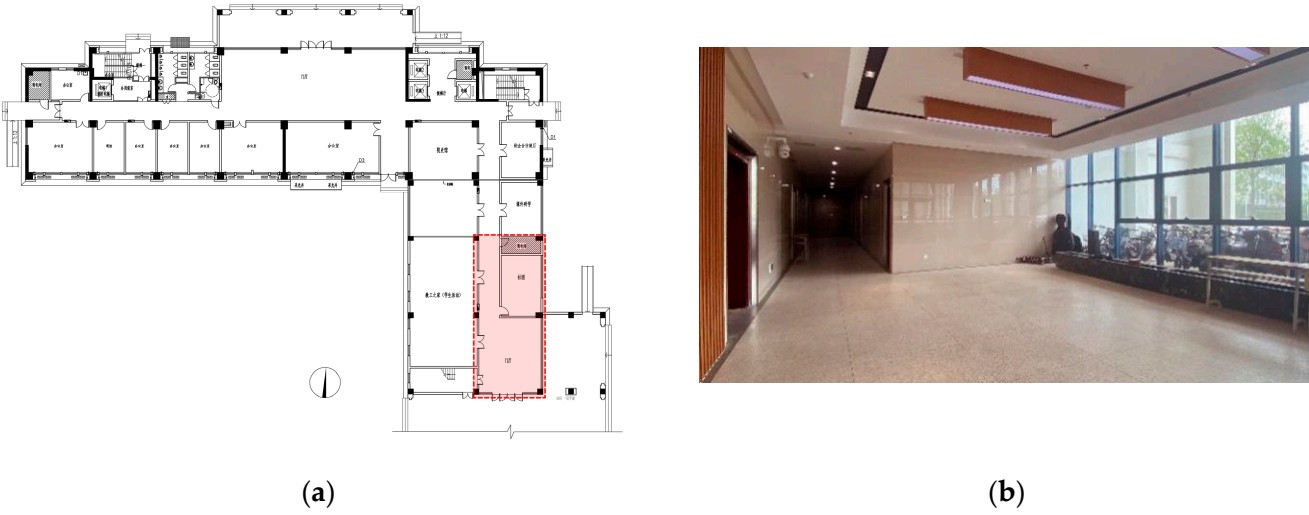

（**a**）　　　　　　　　　　　　　　　　　　　　　　　（**b**）

**Figure 3.** The first-floor cafe before optimization of t. (**a**) Original plan. (**b**) Photo.

(2)　The third-floor terrace (Public/Alone):

As shown in Figure 4, The third-floor terrace is located adjacent to the public rest space and is one of the few spaces in the building where faculty and students can have access to the landscape. However, it was shown in the investigation that there are many problems with this space, such as no facilities for people to rest, and not being open enough to the adjacent spaces. So, students seldom stay there.

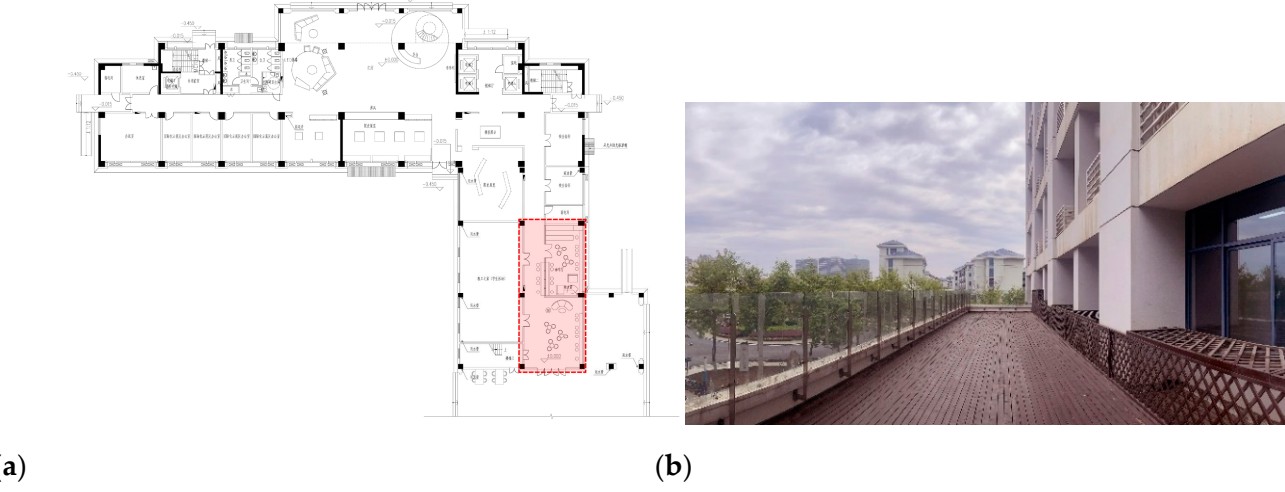

（**a**）　　　　　　　　　　　　　　　　　　　　　（**b**）

**Figure 4.** The third-floor terrace before optimization. (**a**) Original plan. (**b**) Photo.

2.2.2. Optimal Design

Based on the research on the elements of ILS and the problems found in the field research, optimal designs in two case studies were made from the following three aspects: physical space, facilities, and environment. The aim of optimal design was to solve the existing problems and promote the usage by faculty and students.

### 2.3. Visual Perception Experiment

Aim for exploring a much more efficient optimal design method, eye-tracking data from images of the space before and after the optimal design were obtained by visual perception experiment. Being able to capture the visual dynamics of the subjects, the eye-tracking device was used as the main tool for data collection in the experiment. Then the quantified data were analyzed to reflect the real feelings of the subjects in space and thus to check the effectiveness of the optimal design.

#### 2.3.1. Data Collection and Processing

(1) Collecting the typical images before and after optimal design in the case studies.
(2) Making visual perception experiments by utilizing Tobii Pro Lab for quantitative analysis of the perception data of subjects before and after optimal design. Enough valid visual perception data are collected with a wearable eye-tracker and various indexes are selected.
(3) Comparing the visualization results before and after the optimal design by the processing and analysis of eye-tracking data.

#### 2.3.2. Visual Perception Experiment

(1) Subject setting: refer to previous relevant experiment procedures, this experiment was performed a total of 18 times. According to the criteria with a sampling rate greater than or equal to 80% [24], finally 11 sets of valid data were obtained from six subjects.
(2) Stimulus material: images of ILS before and after optimization.
(3) Experiment procedure:

Step 1: once the subject entered the digital laboratory and took their seat, they put on Tobii Glasses 2 (Figure 5a) and performed the first eye-tracking device calibration under the experimenter's instruction.

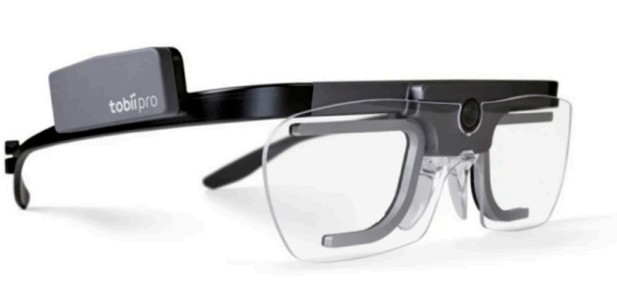
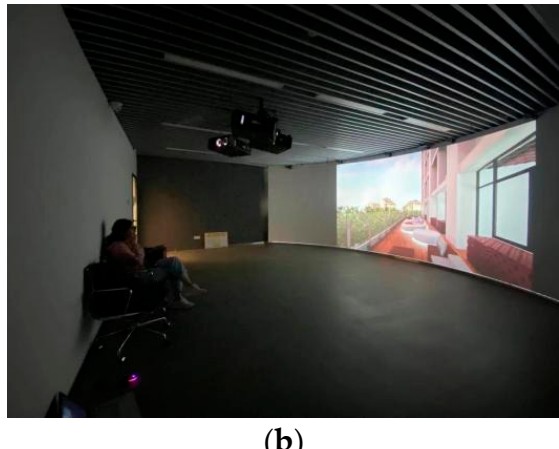

(**a**)                                 (**b**)

**Figure 5.** Visual perception experiment. (**a**) Wearable eye-tracker Tobii Glasses 2. (**b**) Experiment in the digital laboratory.

Step 2: after successful calibration, the experimenter provided the subject a cue and started playing the stimulus material, which the subject watched for one minute.

Step 3: after the first viewing, the experimenter ended the data recording. The subject then proceeded to the second and third viewing under the experimenter's instructions.

Step 4: after the third viewing, the subject removed the oculomotor and left the laboratory.

#### 2.3.3. Eye-Tracking Index Selection

Researchers used the Tobii Pro Lab to make quantitative analysis according to eye-tracking distribution maps, the heat maps and gaze plots, in which the following eye-tracking index was chosen, are shown in Figure 6. In addition, various spatial areas were

defined in both photos and perspectives before and after optimal design, which contributed to creating the areas of interest (AOI). Then, after drawing the target area of interest when analyzing the stimulus material, the following eye-tracking indexes were selected for visual observation: total fixation duration (TFD), the average fixation duration (AFD), the time to first fixation (TFF), the total glance duration (TGD), the average glance duration (AGD), and the glance count (GC) as shown in Table 1. With reference to the study of human visual perception when reading, the six indexes were classified into "visual observation "(TFD/AFD/TFF) and "visual search" (TGD/AGD/GC).

**Table 1.** Eye-tracking indicators and meanings [1].

| Eye-Tracking Concepts and Indicators | | Indicator Abbreviations | Definition | Meaning |
|---|---|---|---|---|
| Visual observation | The total fixation duration | TFD | Total time spent gazing within a given AOI | The longer the time, the more attractive the AOI is |
| | The average fixation duration | AFD | The average duration of all gazes within a given AOI | The longer the time, the more informative or difficult the AOI is to understand, or the more attractive the AOI is |
| | The time to the first fixation | TFF | The time from the beginning of a time interval to a fixation on the first AOI | The shorter the time, the easier the AOI is to engage the subject at the beginning |
| Visual search | The total glance duration | TGD | The duration between the end of the last fixation before entering an AOI (including the entry saccade [2]) and the end of the last fixation within that AOI (before the exit saccade [3]) | The longer the time, the richer the information in the AOI is, the more difficult it is to be understood, or the higher the relevance of all parts in the AOI is |
| | The average Glance Duration | AGD | The average duration of all sweeps within a given AOI | The longer the time, the more invalid the search of the AOI is |
| | The glance count | GC | Number of sweeps for an AOI occurring in a time interval | The higher the number, the longer the subject's search process on that AOI |

[1] Tobii Pro Glasses 2 Product Description (WWW Document), n.d. URL https://nbtltd.com/wp-content/uploads/2018/05/tobiiproproductdescription.pdf, accessed on 1 April 2022; [2] Saccade: central concave vision moves rapidly from one point to another in the oculomotor area. [3] Entry Saccade: the eye movement before the first gaze point in the target AOI. Exit saccade: the eye movement after the last gaze point in the target AOI.

## 3. Results and Discussion

### 3.1. Optimal Design

3.1.1. Case 1: The First-Floor Cafe (Public/Shared)

The optimal design was made according to the three spatial elements. As shown in Figure 6. Firstly, for the physical space, the solid wall of the student activity center was changed to transparent glass to make the space brighter and create multi-level visual communications. While the sub-foyer took utmost advantage of the original windows, the materials used in this space were mainly wood to make people feel much warmer. Second, in terms of facilities, furniture such as chairs, seats, and drinking fountains were added to meet faculty and students' needs for seating and communication and thus attracted them to stay for group learning. Third, for environment, artificial light was added at the top to ensure the quality of the light environment.

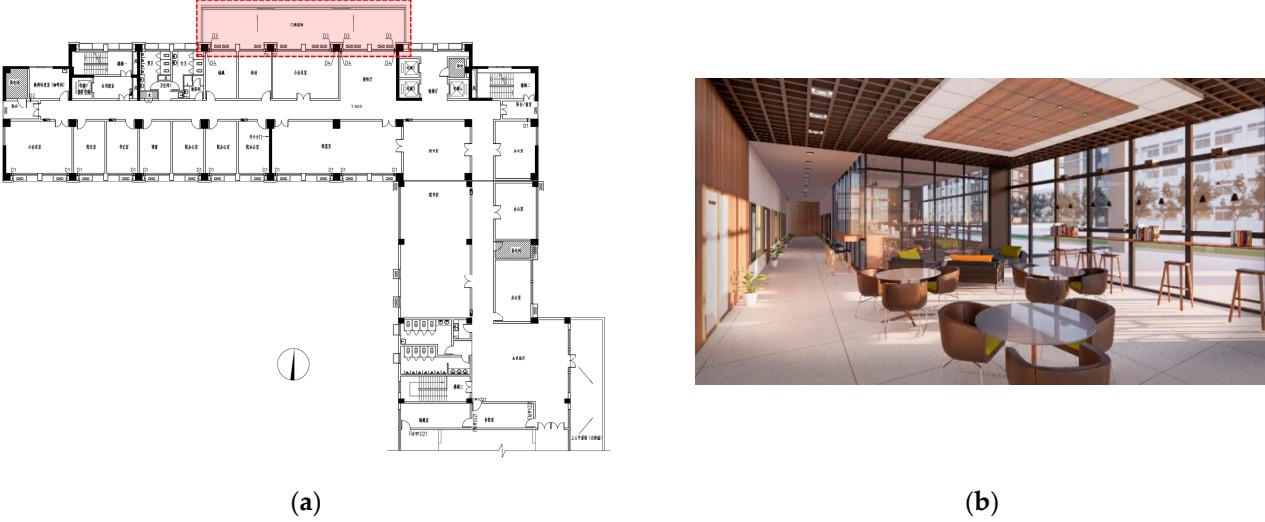

(**a**)  (**b**)

**Figure 6.** The first-floor cafe after optimal design. (**a**) Optimized plan. (**b**) Rendering of the space after optimal design.

### 3.1.2. Case 2: The Third-Floor Outdoor Terrace (Public/Alone)

The optimal design was also made according to the three spatial elements. In terms of the physical space, as shown in the red area in Figure 7, the addition of the door ensured direct access from the rest area to the terrace. As for the facilities, various types of seats were added, and the materials of which were chosen according to the exterior environment. As for the environment, some seats were integrated with artificial light and greenery, which not only enhanced the comfort of users but also offered nighttime lighting.

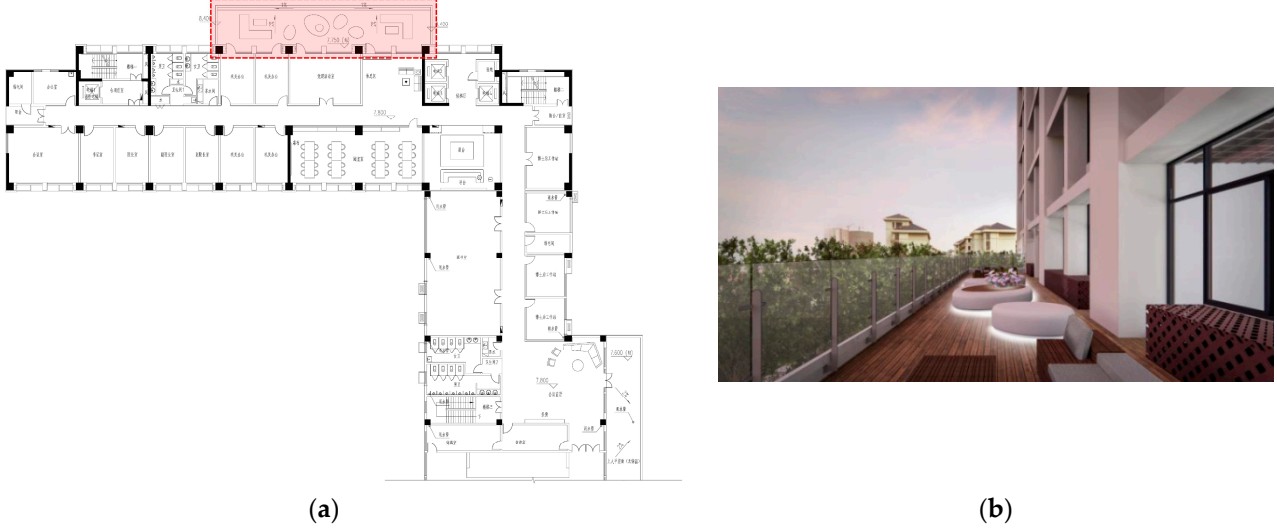

(**a**)  (**b**)

**Figure 7.** The third-floor terrace after optimal design. (**a**) Optimized plan. (**b**) Rendering of the space after optimal design.

### 3.2. Visual Perception Analysis and Verification

#### 3.2.1. Case 1: The First-Floor Cafe (Public/Shared) for Open Group Learning

A comparison of the eye-tracking heat maps before and after optimization in Figure 8a,b showed that the size and number of red and yellow parts were much larger and more numerous than before. The green parts were dispersed over the entire picture, even in the black area outside the image before optimization; shifting to be more convergent, it was mainly concentrated in the horizontal direction after optimal design. This means that the

number of viewers' visual attention points in the first-floor cafe increased after optimization, the visual order was no longer aimless, which demonstrated the space became more attractive after optimal design.

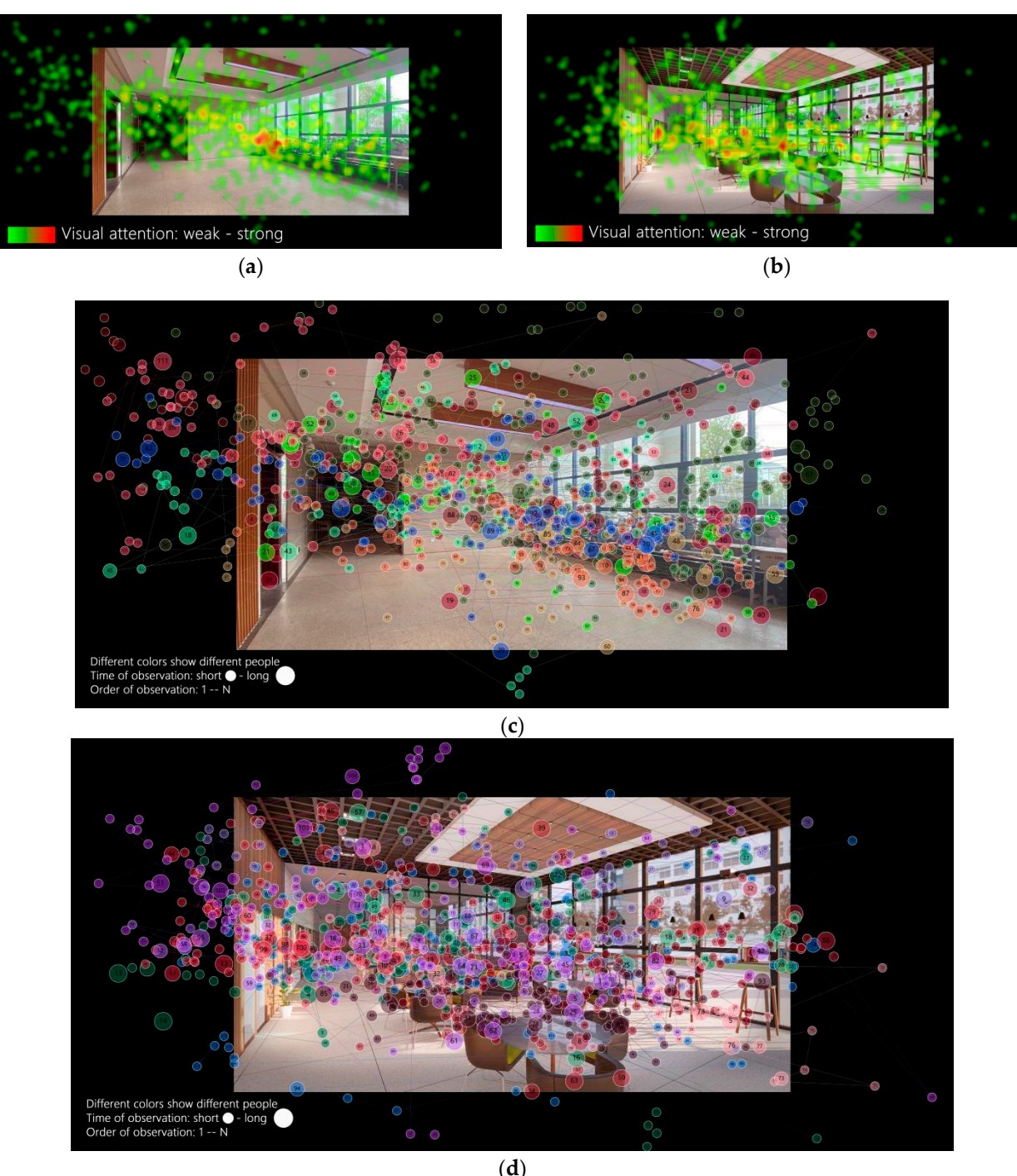

**Figure 8.** Distribution of eye-tracking data of the first-floor cafe. (**a**) The heat map before optimization. (**b**) The heat map after optimization. (**c**) The gaze plots before optimization. (**d**) The gaze plots after optimization.

A comparison of the gaze plots in Figure 8c,d indicated the viewers' visual order. For instance, most of the dots were concentrated on furniture after optimization rather than scattered on every object. Before optimization, the eyesight initially started from the objects in the space, and facade 3, then moved to other spatial elements, and finally returned

to the objects. After optimization, people's eyesight mostly started from furniture 1 and continued to stay on the various facilities. Larger and more widely distributed dots indicate that the optimized space had more ability in attracting people.

The classification of the AOI was shown in Figure 9. The corresponding eye-tracking data were then derived, and a comparison between the spatial elements before and after optimization can be obtained in Table 2.

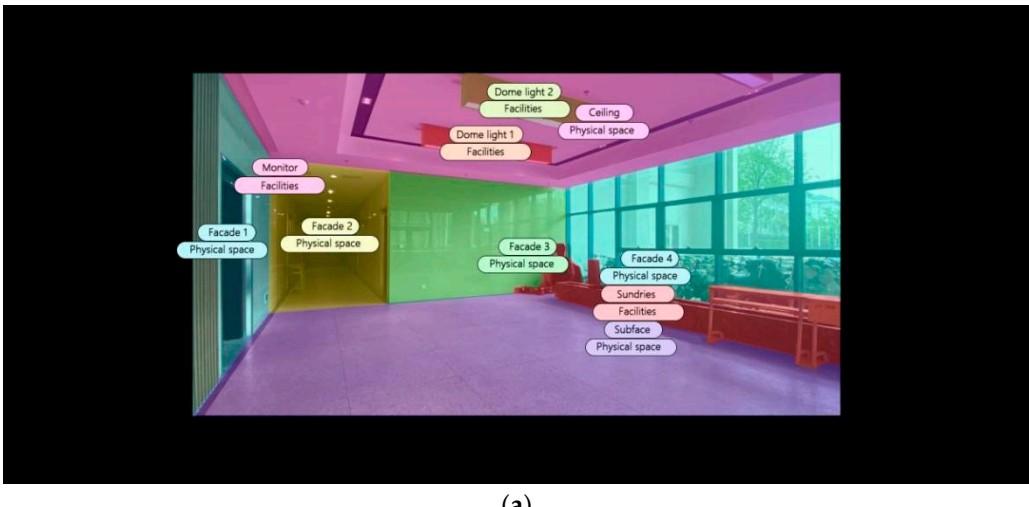

(**a**)

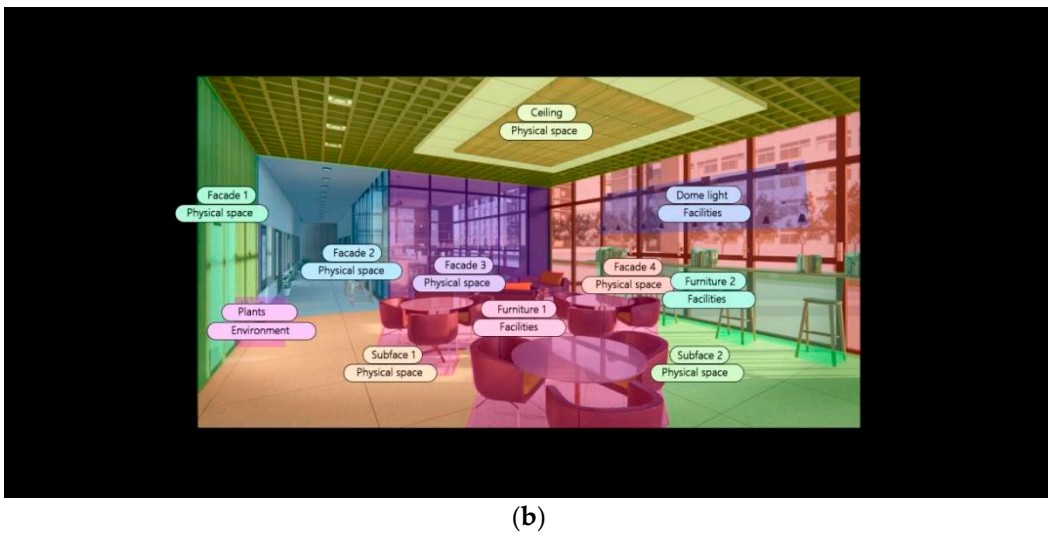

(**b**)

**Figure 9.** AOI division before and after optimization of the first-floor cafe. (**a**) Before optimization. (**b**) After optimization.

(1)    Visual observation (TFD, AFD, and TFF)

In terms of the total index, the TFD of facilities after optimization was higher than before. It was proved that increasing the number of furniture (tables and chairs) can provide people a clear understanding of the spatial function and thus strengthen the recognition of the optimized space.

As for the average index, the AFD of facade 2 and facade 3, which were the main elements of the space, was higher after the optimal design. It meant that variation in the transparency of the facade by changing materials can contribute to building a more attractive space. Moreover, the AFD of the facilities was lower after the optimal design, which meant that the optimized facilities can efficiently convey the functional information of the space.

When it came to the observation sequence, the TFF of facade 1, the top, and the bottom of the physical space increased after the optimal design while the TFF of the facilities decreased. This proved that after the optimal design of the materials of the spatial facade, users paid less attention to the spatial elements that were not directly related to their activities.

**Table 2.** Comparison of eye-tracking data of the first-floor cafe before and after optimization.

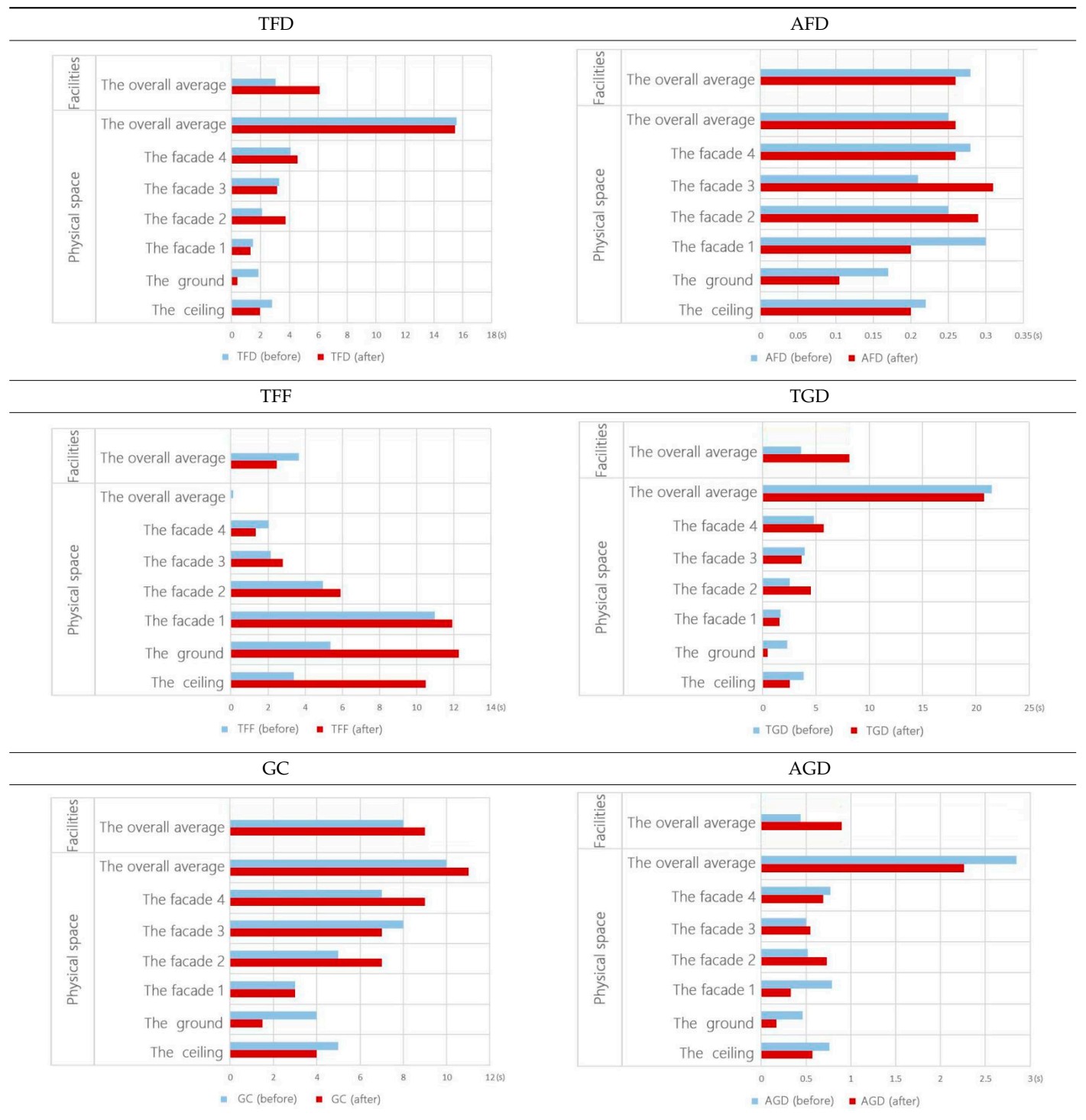

(2)    Visual search (TGD, GC, and AGD)

After optimization, it appeared that the TGD and the GC staying on facilities were more than those before optimization, with less difference between the TGD of physical space and facilities. This showed that the visual attention and level of each component of the space were almost the same, leading to the unity of the space.

As for the average index, the result indicated the difference between the AGD of the physical space and the space facilities shrank after optimization. Thus, it demonstrated that increasing the furniture types can enrich the space hierarchy, helping with the understanding of the space, and reducing the abruptness and discordance in the space. A comparison of eye-tracking data of the first-floor cafe before and after optimization is shown in Table 2.

### 3.2.2. Case 2: Public/Alone: Learning Alone in the Crowd: The Third-Floor Terrace

As seen in the comparison of the eye-tracking heat maps in Figure 10a,b, the area and amount of red and yellow parts of the diagram showed an obvious increase after the optimization. Whereas visual attention which was previously concentrated in the center of the image now distributed horizontally after optimization. Overall, the amount of visual focus attention points on heat maps for the terrace increased after the optimal design.

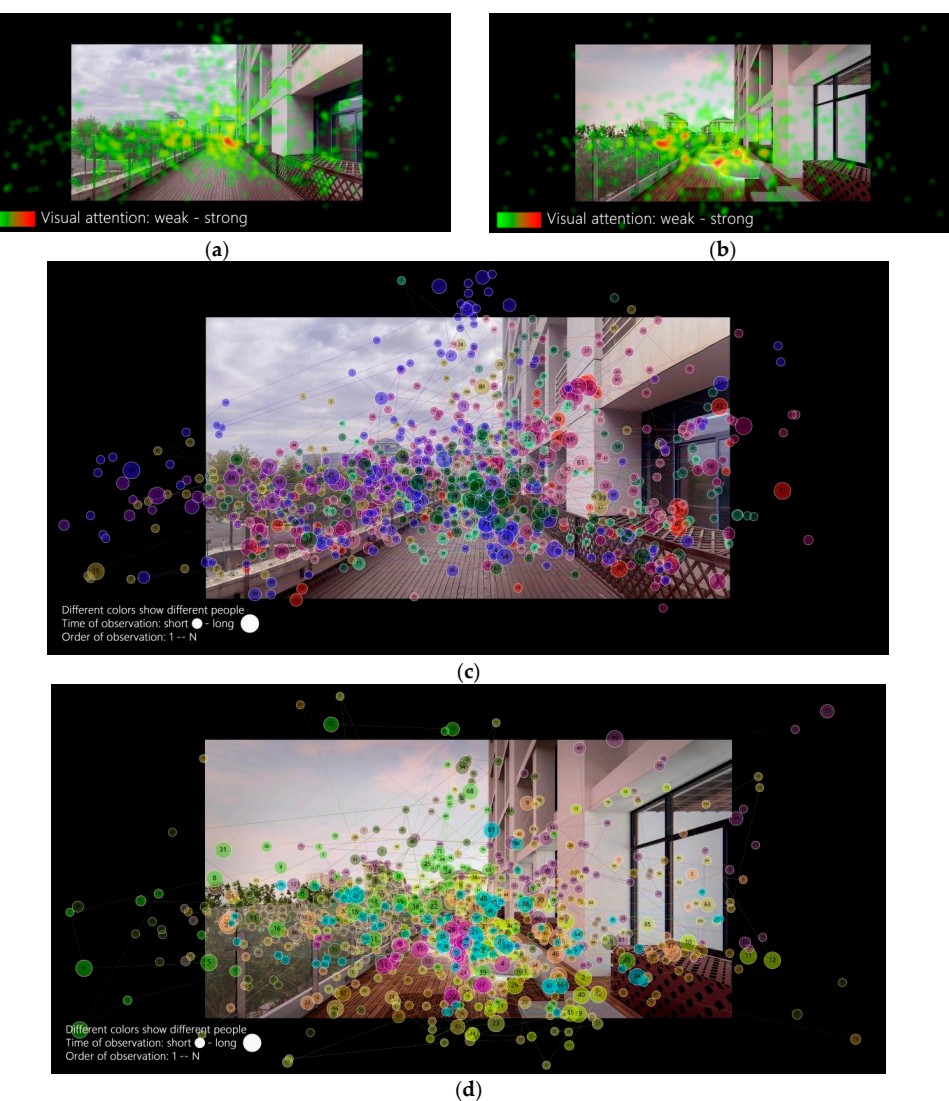

**Figure 10.** Distribution of eye-tracking data of the third-floor terrace. (**a**) The heat map before optimization. (**b**) The heat map after optimization. (**c**) The gaze plots before optimization. (**d**) The gaze plots after optimization.

A comparison of the gaze maps in Figure 10c,d showed that most of the origin points were focused on objects, which represented the users' visual position mostly in the center of the space. The dots were mostly concentrated on the landscape and facilities after optimization, with order of dots indicating the subjects' visual order. As for gaze order, it started from the sky and then shifted to other spatial elements, and finally returned to the facilities repeatedly. The results indicated that the size of the dots is related to the duration of the gaze time. So, it was shown that the optimized exterior space with natural landscape and facilities was more attractive.

The classification of the AOI according to spatial elements is shown in Figure 11. Eye-tracking data were then derived, and a comparison between the spatial elements before and after optimization can be seen in Table 3.

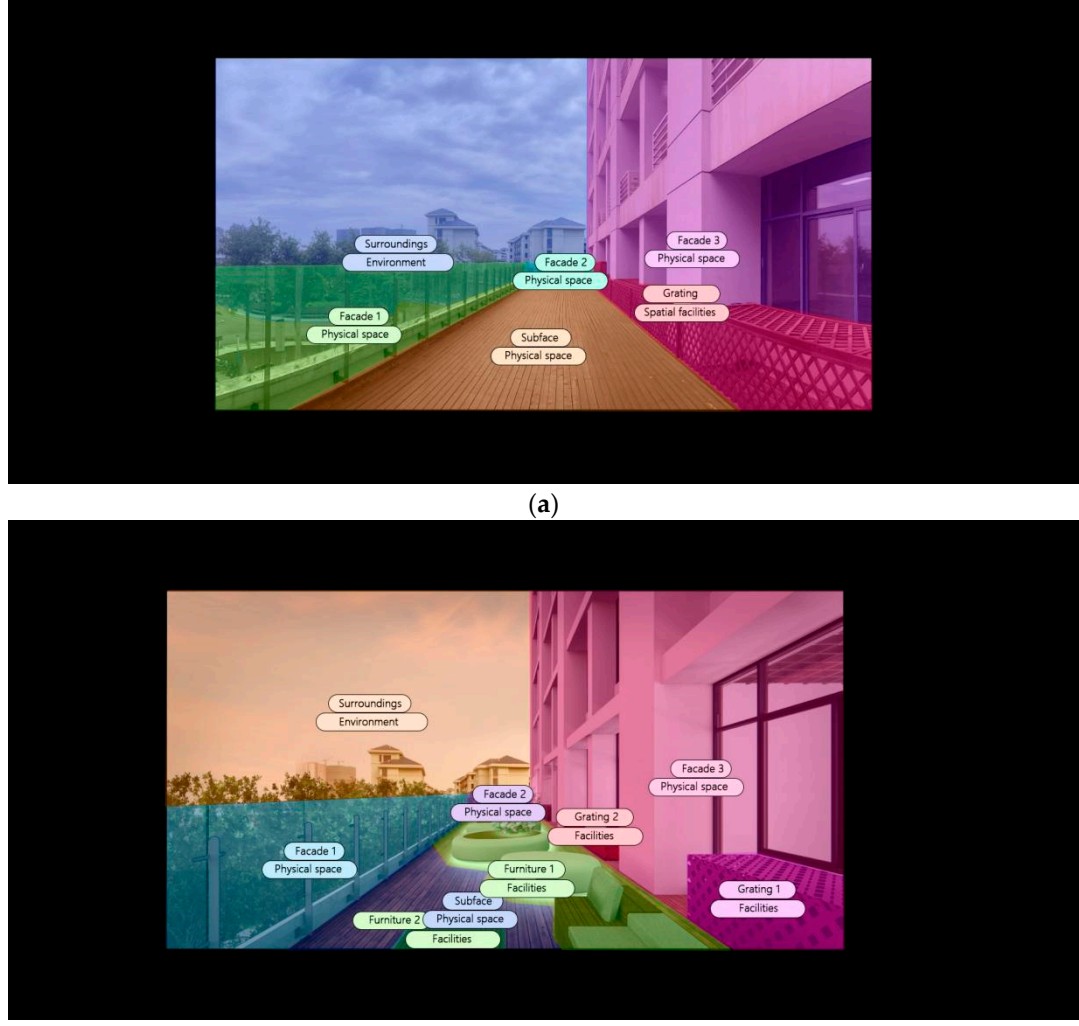

(**a**)

(**b**)

**Figure 11.** AOI division diagram before and after optimization of the third-floor outdoor terrace. (**a**) Before optimization. (**b**) After optimization.

(1)   Visual observation (TFD, AFD, and TFF)

In terms of summation index, after optimization, the TFD of total observation time of facilities was more than that before optimization, and the TFD of environment and physical space was less than that before optimization. The TFD gap between the three types of material spatial elements was reduced. It proved that after the spatial elements were sorted out, the various spatial elements in this space became more homogeneous.

**Table 3.** Comparison of eye-tracking data of the third-floor terrace before and after optimization.

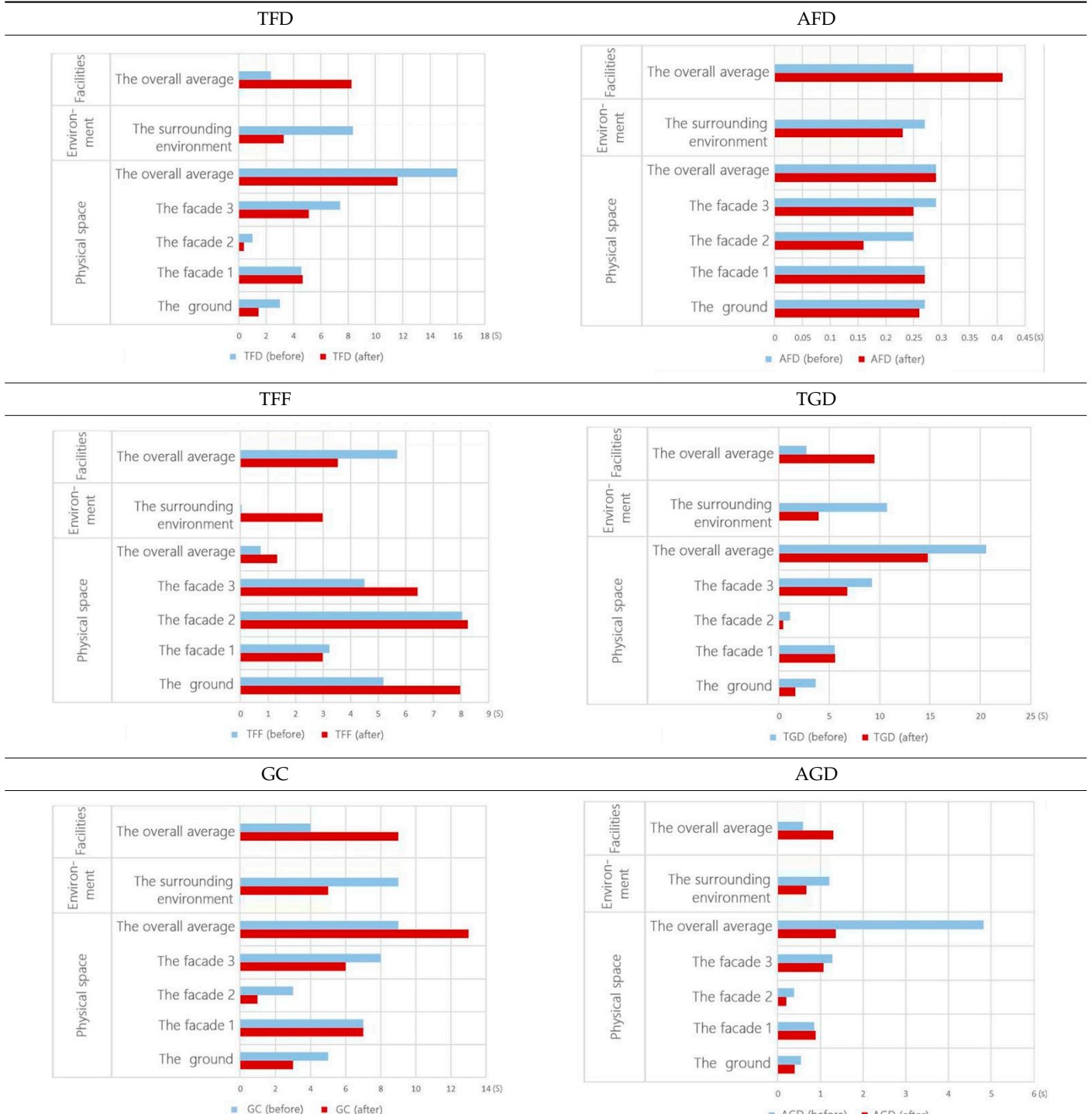

As for the averaging index, the AFD in the facility increased after optimization. It proved that the addition of the seats required for the ILS helped to complete the space.

For the observation sequence, after the optimization, the TFF for the surrounding environment increased. This proved the decrease in the influence of the surrounding environment on users' attention after the optimal design.

(2)    Visual search (TGD, GC, and AGD)

In terms of summation index, after optimization, the TGD and the GC of facilities were more than those before optimization, and the difference among the three spatial factors was reduced. This proved that the integrated design of greenery and seating improved the visibility of facilities and attracts users to stay.

In terms of the averaging index, the difference between the AGD of the physical space, environment, and facilities was reduced after optimization. This proved that the spatial elements work well and created a harmonious and natural atmosphere. A comparison of eye-tracking data of the third-floor terrace before and after optimization is shown in Table 3.

## 4. Conclusions

The ILS in universities is receiving more and more attention as China's higher education enters a new stage with the development model characterized by stock optimization. Based on Weber's four-quadrant theory and field investigation, this paper proposes a revised theoretical research of campus ILS classifications and spatial elements more applicable to the study of architectural spaces. Then, two case studies were made to explore optimal design methods from the three spatial elements of physical space, facilities, and environment. Visual perception experiments using visual perception technologies were made to verify the effect of optimal design by studying the relationship between users' visual perception and spatial elements for further improvements to optimal design methods. This paper draws the following conclusions:

(1)     For the optimization of ILS in universities, it is worth paying attention to the physical space, such as size, enclosure, richness, transparency, and other elements of the space. For example, the facade can be homogenized to lower the impact of visual interference of facade information to users. Removing the unrelating elements on facades can reduce the excessive visual attention of users. Hence, if the facade applies the materials with complicated context, it should be placed in a larger space to make the space more identified. Besides, adjusting the colors and materials on the facade is suitable to establish an active and vivid space. It is also helpful to optimize facilities' number, location, and accessibility. Besides, facilities can be combined with landscape elements to create a natural atmosphere.

(2)     Visual perception experiments and quantitative analysis demonstrate that the quality of ILS after optimization is much improved. Comparisons of both visual observation factors (TFD, AFD, and TFF) and visual search factors (TGD, GC, and AGD) in the two case studies show that optimized space (exterior space with natural landscape) and facilities were more attractive.

Due to resource and time constraints, there are some points in this paper that need to be improved:

(1)     It is also necessary to increase the sample size of subjects and to study two other types of spaces, namely the spaces of Private/Alone (Individual learning), and Private/Shared (Group learning), to further refine the experimental design and overall study content.

(2)     The verifying process can be supplemented with questionnaires and other methods to better understand users' feelings.

(3)     The perceptual data in this study, especially for human visual perception, need more technical means and more types of human perceptual data, such as EEG, to obtain more comprehensive and integrated analysis results.

**Author Contributions:** Conceptualization, J.W. and Y.C.; methodology, Y.C.; software, Y.C.; validation, Y.C.; formal analysis, Y.C.; investigation, Y.C.; resources, J.W. and X.Z.; data curation, Y.C.; writing—original draft preparation, Y.Z.; writing—review and editing, Y.C., W.D. and J.W.; visualization, Y.Z.; supervision, J.W., W.D. and X.Z.; project administration, J.W. and Y.C.; funding acquisition, J.W. All authors have read and agreed to the published version of the manuscript.

**Funding:** This research was funded by RESEARCH PROGRAM OF NATURAL SCIENCE FOUNDATION OF CHINA, grant numbers 52078113, 51678123, and 52078117.

**Institutional Review Board Statement:** Not applicable.

**Informed Consent Statement:** Not applicable.

**Data Availability Statement:** Not applicable.

**Conflicts of Interest:** The authors declare no conflict of interest.

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
