# Peer review of "Optimal Design and Verification of Informal Learning Spaces (ILS) in Chinese Universities Based on Visual Perception Analysis"

_buildings, doi:10.3390/buildings12101495_

Round 1

Reviewer 1 Report

1.         Generally, this paper presents an exciting way to optimize some unattractive spaces by the optimal design methods based on visual perception analysis. It can also be applied to other areas, such as parks, gardens, and squares.

2.         Section 3.2, it looks weak in how the authors obtain the conclusion. The difference between before and after the optimization, Figure 8(a) and 8(b), is not quite significant.

3.         Table 2, The overall average of the Facilities by AFD and TFF show that after optimization is less than before. Could it conclude that the optimization is successful?

4.         The conclusion section seems too long. To be a little bit shortened is recommended.

Reviewer 2 Report

·       The topic of the paper is quite interesting however the paper seems to be in the early stages of development.

·       Full editing and proofreading is required to edit mistakes including the first reference in the References list, for example.

·       The literature review is generally descriptive, and the list of references is rather limited, this should be extended and deeper engagement with previous research in this area is needed. Currently, there’s no section in the paper that critically analyses previous literature. Citing some references in the introduction is not sufficient.

·       The paper has too many aims and no objectives. The paper should have an overall aim which can then be broken down into 3-5 practical steps (objectives) that can help achieve the overall aim. The overall aim can be the first aim mentioned in the paper. The aim should be reflected in the title which is currently appropriate: Optimal design and verification of Informal Learning Spaces (ILS) in Chinese universities based on visual perception analysis.

·       The aim and objectives can be along the lines of:

Aim: To investigate optimal design and verification of ILS in Chinese universities using visual perception analysis to solve existing problems and promote usage by both faculty and staff.

 Objectives:

1.       To verify the effect of optimal design of ILS on visual perception experiment and quantitative analysis.

2.       To study the relationship between users’ visual perception and spatial elements for further of optimal design methods

3.       To propose practical recommendations to enhance the use of ISL in Chinese universities

·       The contribution of this research to the wider bodies of literature is underrepresented. The impact of this research to theory, practice, policy making etc. must be clarified.

·       There are major shortcomings in the methodology section. The authors need to clarify the overall approach to this research/methodology from the outset. Check Research Design and Methods textbooks and papers for guidance. The methodology section heavily focuses on the experiment/procedure in the current draft which is not enough. References related to the methodology must be cited in this section. The authors have to justify why this methodology was chosen and why is it the best fit to help achieve the aim and objectives of this research. Evaluate if similar studies have used the same approach and insert some relevant citations.

·       The study is using one university campus only, why was this case study chosen and is it representative to Chinese universities?

·       Why was Scott-Webber’s (2015) framework used? It should be introduced and discussed properly. How did it inform your study?

·       The data sample (6 people and 11 sets of data) is very small. The authors need to either justify how this is sufficient/representative or collect more data. It is not sufficient to mention this in the conclusions as limitations of this study related to time and resources constraints.

·       It is not clear what the Meanings/interpretations in Table 1 were based on, explain.

·       The findings (section 3.2) seem to be obvious and expected. The authors need to articulate the unique contribution of this paper and how are findings are compared/contrasted with previous literature discussed earlier in the paper to clarify the positioning of this research within the wider bodies of literature and the contribution to knowledge.

·       The two conclusions in section 4 are repetitive as they were mentioned in section 3.3, delete one of them.

Round 2

Reviewer 2 Report

The revised paper has improved however, points 2, 5, 6  & were not fully addressed. The authors need properly engage with previous literature, particularly from 'Buildings' journal to properly position their research within relevant bodies of literature and sharpen the contribution of this research. The literature review is still limited, adding a few citations is not sufficient.

Round 3

Reviewer 2 Report

The paper has improved but the literature review is descriptive. The authors claim to identify the strengths and weaknesses of previous studies but this is not demonstrated clearly.
